# Discomfort and Pain Related to Protective Mask-Wearing during COVID-19 Pandemic

**DOI:** 10.3390/jpm12091443

**Published:** 2022-09-01

**Authors:** Luca Padua, Letizia Castelli, Dario M. Gatto, Keichii Hokkoku, Giuseppe Reale, Roberta Pastorino, Claudia Loreti, Silvia Giovannini

**Affiliations:** 1Department of Geriatrics and Orthopaedics, Università Cattolica del Sacro Cuore, 00168 Rome, Italy; 2High Intensity Neurorehabilitation Unit, Fondazione Policlinico Universitario A. Gemelli IRCCS, 00168 Rome, Italy; 3Department of Aging, Neurological, Orthopaedic and Head-Neck Sciences, Fondazione Policlinico Universitario A. Gemelli IRCCS, 00168 Rome, Italy; 4Department of Neurology, Teikyo University School of Medicine, Tokyo 173-8605, Japan; 5Department of Neurosciences, Università Cattolica del Sacro Cuore, 00168 Rome, Italy; 6Department of Woman and Child Health and Public Health—Public Health Area, Fondazione Policlinico Universitario A. Gemelli IRCCS, 00168 Rome, Italy; 7Hygene Section, Department of Life Sciences and Public Health, Università Cattolica del Sacro Cuore, 00168 Rome, Italy; 8Post-Acute Rehabilitation Unit, Fondazione Policlinico Universitario A. Gemelli IRCCS, 00168 Rome, Italy

**Keywords:** COVID-19, facemask, pain, neuropathic pain, personalized medicine

## Abstract

The SARS-CoV-2 pandemic made the use of facemasks mandatory to prevent contact with the virus. Recent studies have revealed that intensive use of facemasks significantly exacerbated pre-existing headaches and triggered de novo headaches. In our experience, some subjects also complain of symptoms of neuropathic pain in the head/facial regions. Until now, the relationship between neuropathic pain and facemasks has not been documented. The aim of the study is to investigate the occurrence of neuropathic pain related to facemask use. It is a cross-sectional survey using a questionnaire, developed following a commonly accepted outcome research methodology. Participants, both health care and non-health care workers, responded to items included in the questionnaire about the type of facemasks, time and manner of wearing them, side effects such as skin lesions, symptoms of neuropathic pain, etc.

## 1. Introduction

The dramatic spread of COronaVIrus Disease 2019 (COVID-19), caused by Severe Acute Respiratory Syndrome CoronaVirus 2 (SARS-CoV-2), has imposed international measures that mandate personal protective equipment (PPE), in particular facemasks, that together with social and physical distancing, is considered the most useful way to prevent contact with the virus [1,2].

While the scientific community acknowledges the benefit of PPE to prevent the viral infection, some concern has been raised regarding the adverse effects of PPE [3,4]. Most of the recent studies revealed that intensive usage of facemasks, which is the most used PPE, greatly exacerbated preexisting headaches and even triggered de novo headaches [5,6,7,8,9,10,11], even before the COVID-19 pandemic [5,11].

The International Association for the Study of Pain first introduced the term “neuropathic pain” defined as “pain initiated or caused by a primary lesion or dysfunction in the nervous system” [12]. This definition was useful for discriminating neuropathic pain from other type of pain, but the more precise re-definition proposed by Treed and colleagues: “pain arising as a direct consequence of a lesion or disease affecting the somatosensory system” will be helpful for clinical and research purposes and fits into the nosology of neurologic disorders [13]. Neuropathic pain differs from nociceptive pain in terms of pain perception [14,15,16].

Pain is a personal experience, and for headaches, several subjects complain of pain in the head and facial regions that, due to the features, could be defined as neuropathic pain. To describe the pain, subjects often used descriptions such as “altered sensitivity”, “numbness”, “electric shock-like pain”, “tingling”, “pinprick”, “burning”, “itching”, etc. They usually related those symptoms to the wearing of facemasks. The relationship between neuropathic pain and PPE has not been well investigated or documented. 

The present study aimed to investigate the frequencies and features of neuropathic pain in the head and facial regions related to facemask usage by an ad hoc personalized questionnaire (pain is usually studied by self-reported scales and questionnaires) [14,17,18,19].

## 2. Materials and Methods

We performed a cross-sectional survey to assess the frequency of symptoms suggesting neuropathic pain in a large sample through an ad hoc developed self-administered questionnaire. The ad hoc questionnaire was developed according to the accepted methodology [20,21]. The first phase was the “item generation” which involved asking a sample of persons working in and out of the hospital to propose questions assessing the pain and features of the pain occurring at the head and facial region related to wearing facemasks. This methodology warranted a questionnaire developed through several perspectives (different kinds of workers, ways of using the facemasks, risks of exposure to the virus, etc.). After item generation, a second phase of “item reduction” was done by merging similar questions and erasing those that were redundant. The items were then evaluated by physicians skilled in neuropathic pain to add, if necessary, the lexical attributes typical of neuropathic pain. The final version of the questionnaire was a self-administered 19-item multiple-choice questionnaire including items on age, gender, job, number of hours wearing PPE, type of facemask, and how the facemask is worn (e.g., wearing an “ear protectors” such as cotton ties, adjustable extension cords, or other). Concerning pain, the questionnaires included items focused on assessing, through verbal descriptors, if subjects complained of transitory episodes of typical neuropathic pain symptoms (electric shock-like pain, tingling, burning, pinprick, itching, etc.) [14,22,23] and if those symptoms were associated with reduced skin sensitivity (hypoesthesia, numbness, anesthesia, etc.), which is a further stage of neuropathic involvement. After checking that the time to fill in the last version of the questionnaire was appropriate and that all the items were comprehensible, a cross-sectional survey was conducted during March and April 2021 through Google Forms. The questionnaire was sent to a widespread mailing list and social network without filtering the participants (English translated version is available as a Appendix A).

Each participant gave his or her written consent to participate in the study, and his or her data were processed anonymously.

At the end of the period, data from the Google spreadsheet were analyzed using SPSS (IBM SPSS Statistics for Windows, Version 25.0. IBM Corp, Armonk, NY, USA); the features of the data variables were evaluated using the chi-square test and Mann–Whitney U test. Statistical significance was defined as a *p*-value < 0.05. This study was conducted in accordance with the International Guidelines for Good Clinical Practice and the Declaration of Helsinki, and all subjects gave written informed consent prior to participation.

## 3. Results

The participants consisted of health care workers taking care of non-COVID-19 subjects in general wards and non-healthcare workers. Six hundred and three participants completed the questionnaire. The sample consisted of 403 females and 200 males. The average was 38.2 years ± 14.6 SD (range 16–88). In terms of job categories, most were paramedics (nurses, physical therapists, 34.5%), followed by doctors (19.6%), office workers (15.6%), students (14.6%), engineers (6.3%), teachers (4.5%), retired/unemployed (1.7%), homemakers (1.5%), pharmacists/biologists (1.3%), and military officers (0.5%).

Based on our data, the most used facemask was FFP2 (65.2%), followed by surgical masks (33.6%), and finally FFP3 (1.2%). The mean time of facemask daily usage was 7 h ± 3.12 SD (range 1–16; median: 8); 80.8% of participants wore a single facemask, while 18.2% wore a double facemask. About a fifth (19.6%) of the entire sample reported that they had to change the type of facemask because of skin lesions. 

Most participants (88.6%) reported wearing facemasks with elastic bands behind their ears, while 11.4% reported using “ear-savers”, i.e., alternative strategies to protect their ears from skin lesions. Regarding the skin lesions: 50% had appeared in the upper ear, 43.8% in the middle part of the ear, and 6.2% in the lower part of the ear, approximately at the lobe junction.

Ear-saver strategies used were: 41.2% adjustable extensions or cords, 39.7% plastic mask band, 7.3% cotton ties, 7.3% cotton cap with side buttons, and 4.5% elastic bands. Considering the population using ear-savers, 30.9% wore them in the up position (between parietal an occipital areas), 26.5% in the mid-head (about halfway up the occipital area), 22.1% down about neck level, while 20.6% said they changed positions frequently to avoid “stressing” one position more than another.

### 3.1. Neuropathic Pain

Around 1 out of 5 participants (19.2%, *n* = 116) presented head and/or facial symptoms suggesting neuropathic pain (Table 1). 

The relationship between the type of facemask and the occurrence of neuropathic pain was not significant (*p* = 0.732). However, the relationship with the time of wearing facemasks was significant: subjects complaining of symptoms suggesting neuropathic pain reported wearing facemasks for a longer time compared to subjects not complaining of symptoms (subjects with neuropathic symptoms 7.6 ± 2.8 h, subjects without neuropathic symptoms hours 6.8 ± 2.9, *p* = 0.007). We also observed a strong association between changing the type of facemask and neuropathic pain symptoms (*p* < 0.001). Among the group of subjects who reported neuropathic pain symptoms, 15% (*n* = 17) also reported reduced skin sensitivity (2.8% of the entire sample). We also observed a strong association between changing the type of facemask and reduced skin sensitivity (*p* < 0.001). The relationships between the occurrence of reduced skin sensitivity and the type of facemask and time of wearing were not significant. 

### 3.2. Other Discomfort

Overall, 24.2% of the sample reported having more headaches than usual, while 17.7% reported having de novo headaches (25% reported not experiencing headaches, and 33% reported no worsening in the number of headaches). For those who experienced de novo or a worsening headache, 75% suffered from band-like headaches, while the remaining 25% suffered from throbbing headaches. Participants who reported symptoms suggesting neuropathic pain (§ 3.1.; *n* = 116) reported more headaches than usual in 39.6% of cases (*n* = 46) and de novo headaches in 22.4% (*n* = 26); 27.6% of them (*n* = 32) reported no differences, and 10.4% reported not suffering from headaches.

The questionnaire included an open question on other disorders/discomfort that the subjects related to wearing protective facemasks. Three hundred and sixty-one participants (59.9%) reported that wearing facemasks daily caused some other discomfort or disorder. Half of them (49.9%) reported that wearing a face mask caused skin problems such as acne and irritations, 21.3% reported cervical pain, 12.5% bruxism and temporomandibular occlusion, and 3.6% reported some respiratory disorder. Table 2 shows other disorders that emerged from the questionnaire and their occurrence. Note that a positive association was observed between neuropathic pain symptoms and acne/skin irritation (*p* = 0.012).

## 4. Discussion

Due to the pandemic caused by COVID-19, mask use has become part of everyday life [1,24,25]. In the literature, there are studies that have investigated the relationship between the use of facemasks and the presence of acne, skin lesions [26], and headaches [5,6,7]. However, there is no work that has investigated the presence of neuropathic pain: a disabling condition that impacts on daily activities and on quality of life [16,27,28].

Our survey found that 19.2% of participants reported suffering neuropathic pain symptoms: in half of the cases, it was localized to the nasal region, followed by pain in the zygomatic, parathyroid, temporal, occipital, and submental regions, all of which were areas of contact with facemasks [6]. It is also relevant to point out that 15% of the population complaining of neuropathic pain also reported suffering from reduced skin sensitivity, which is a further stage of nerve involvement. The localization of most symptoms suggests the involvement of the trigeminal nerve, particularly the maxillary and mandibular branches, and usually the localization of the pain may be attributed to the involvement of single, small, or terminal trigeminal branches [29,30].

Around 1 out of 5 participants complained of skin lesions requiring them to change the type or modality of wearing facemasks. From the analysis, it emerged that most of the subjects who changed the type of facemask were also those who complained of neuropathic pain symptoms and reduced skin sensitivity, and therefore, it is likely that, the extent of the neuropathic pain was a further reason to change the facemask use approach (change the type or modality of wearing facemask).

The probable mechanisms of the nerve involvement would be the prolonged compression, even if mild, of a nerve branch (often a small/terminal one) by the facemask. The need to have the mask close to the face (almost hermetic) to protect the respiratory tract and reduce the exposure to the virus, likely cause a prolonged compression of the facemask on the skin and on the underlying nerve branches. The results show that the longer a facemask is worn, the higher the probability of neuropathic pain, which confirms the hypothesis. According to previous data reported in the literature, most participants suffered from acne and/or skin irritation, neck pain, and bruxism [26]. These problems were followed by a miscellany of complaints, ranging from ocular, gastrointestinal, fatigue, and dizziness to psychological disorders. Note that a positive association was observed between neuropathic pain symptoms and acne/skin irritation; this is probably due to skin inflammation involving the intradermic terminal nerve fibers. Further studies should shed light on the association between neuropathic pain and a previous history of headaches (especially migraine and tension type headache) or other facial pain syndromes: allodynia, for example, is a common symptom in patients with migraine.

## 5. Conclusions

In conclusion, neuropathic pain is very subjective and should be recognized as a facemask-related adverse event. Although there is no established preventive approach for these complications, appropriate education on mask use should be implemented such as avoiding wearing them continuously in the same modality and ensuring a proper fit. Finally, if these data are confirmed and the pandemic condition requires face masks to be worn for a long time to come, the modification of facemasks should be considered.

## Figures and Tables

**Table 1 jpm-12-01443-t001:** Characteristics of subjects and localization of head and/or facial neuropathic pain symptoms.

	N = 116	
**Gender**	F:M	73:43
**Age**	Mean ± SD	35.5 ± 12.9
**Employment**		
Paramedics/Healthcare		42.30%
Doctors		15.50%
Office workers		19.00%
Student		12.10%
Engineer		6.90%
Teacher/Professor		3.40%
Pharmacist/biologist		0.80%
**Mask type**		
FFP2		69.80%
Surgical		27.70%
Cotton surgical		1.70%
FFP3		0.80%
**Localization**		
Nasal region		50.90%
Zygomatic region		30.20%
Parotid region		19.80%
Temporal region		13.80%
Occipital region		12.90%
Submental region		12.90%
Frontal region		9.50%
Lips/buccal region		7.80%
Mastoid region		5.20%
Orbital region		2.60%
Supraorbital region		1.70%
Scalp region		0.90%

**Table 2 jpm-12-01443-t002:** Other discomfort resulting from the use of the mask.

Types of Disorders (N = 361)	
Acne/skin irritations	49.9%
Cervical pain	21.3%
Bruxism and temporomandibular occlusion	12.5%
Respiratory disorders	3.6%
Retro-auricular pain	2.5%
Eye discomfort (e.g., irritation, dryness)	2.5%
Itching, erythema, beard alopecia	1.9%
Drowsiness, fatigue, dizziness	1.4%
Herpes labialis	1.1%
Excessive salivation and reflux	0.5%
Dysphonia, communication difficulties	0.4%
Psychological disorders	0.4%

## Data Availability

The data presented in this study are available on request from the first author.

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
