# Peer review of "Discomfort and Pain Related to Protective Mask-Wearing during COVID-19 Pandemic"

_jpm, 2022, doi:10.3390/jpm12091443_

Round 1

Reviewer 1 Report

The large number of medical personnel involved and the evaluation of the use of protective masks should be appreciated.

For doctors and nurses, the use of this equipment is nothing new, but the Sars-Cov-2 pandemic has determined the use of the mask for a long time.

In the questionnaire used, there is only question 16 (16. Using the mask, did you experience a burning sensation, electric shock, pinprick and/or tingling?) which refers to symptoms associated with neuropathic pain that can be answered with yes or no. I don't consider just one question enough to consider neuropathic pain. For the diagnosis of neuropathic pain, questionnaires or neuropathy assessment scales must be used. And from a statistical point of view, no significant results appear. 

I suggest changing the title of the article, neuropathic pain is not sufficiently evaluated by the questionnaire questions.

Author Response

We would like to thank the reviewer for his/her comments. We have changed the title following the suggestion.

We hope the manuscript is now suitable for publication in JPM.

Reviewer 2 Report

Thank you for your nice survey. I find it interesting and scientific significant. Only a view comments:

L 45-46: Reference 5, and Ref 11, the surveys are made BEFORE the COVID-19 pandemic. Maybe you could highlight that.

L58 (and L82):  neuropathic pain can also be itching” in one-third of cases, Hachisuka et al., Pain 2018

L83: Did you also asked for “allodynia”?

L101: the male participants were the half of the females. Did you see any difference in pain localization between Gender?

Generally: it would be very interesting to know how many of the 116 participants with neuropathic pain had a previous history with headaches (especially migraine, but also tension type headache) or other facial pain syndroms. Allodynia (I mention before), for example, is common in patients with migraine.

Author Response

We would like to thank the reviewer for his/her insightful suggestions that allow us to improve our paper.

L 45-46: Reference 5, and Ref 11, the surveys are made BEFORE the COVID-19 pandemic. Maybe you could highlight that.

Thank You, we have highlighted it as suggested.

L58 (and L82):  neuropathic pain can also be itching” in one-third of cases, Hachisuka et al., Pain 2018

Thank You, we have added it as suggested.

L83: Did you also asked for “allodynia”?

Thank You, we have added this in discussion as suggested.

L101: the male participants were the half of the females. Did you see any difference in pain localization between Gender?

Thank You, no significative differences between gender in pain localization were found

Generally: it would be very interesting to know how many of the 116 participants with neuropathic pain had a previous history with headaches (especially migraine, but also tension type headache) or other facial pain syndroms. Allodynia (I mention before), for example, is common in patients with migraine.

Thank You, we have added a paragraph in results about it and in the discussion for further studies as suggested.